# A Sesquiterpene Synthase from the Endophytic Fungus *Serendipita indica* Catalyzes Formation of Viridiflorol

**DOI:** 10.3390/biom11060898

**Published:** 2021-06-16

**Authors:** Fani Ntana, Wajid W. Bhat, Sean R. Johnson, Hans J. L. Jørgensen, David B. Collinge, Birgit Jensen, Björn Hamberger

**Affiliations:** 1Department of Environmental Science, Aarhus University, Frederiksborgvej 399, 4000 Roskilde, Denmark; fntana@envs.au.dk; 2Department of Biochemistry and Molecular Biology, Michigan State University, 603 Wilson Rd, East Lansing, MI 48824, USA; bhatwaji@msu.edu; 3New England Biolabs, Inc., 240 County Road, Ipswich, MA 01938, USA; sjohnson@neb.com; 4Department of Plant and Environmental Sciences and Copenhagen Plant Science Centre, University of Copenhagen, Thorvaldsensvej 40, 1871 Copenhagen, Denmark; hjo@plen.ku.dk (H.J.L.J.); dbc@plen.ku.dk (D.B.C.); bje@plen.ku.dk (B.J.)

**Keywords:** antifungal, Basidiomycota, endophyte, *Piriformospora indica*, sesquiterpenoid, terpene synthase, viridiflorol

## Abstract

Interactions between plant-associated fungi and their hosts are characterized by a continuous crosstalk of chemical molecules. Specialized metabolites are often produced during these associations and play important roles in the symbiosis between the plant and the fungus, as well as in the establishment of additional interactions between the symbionts and other organisms present in the niche. *Serendipita indica*, a root endophytic fungus from the phylum Basidiomycota, is able to colonize a wide range of plant species, conferring many benefits to its hosts. The genome of *S. indica* possesses only few genes predicted to be involved in specialized metabolite biosynthesis, including a putative terpenoid synthase gene (*SiTPS*). In our experimental setup, *SiTPS* expression was upregulated when the fungus colonized tomato roots compared to its expression in fungal biomass growing on synthetic medium. Heterologous expression of *SiTPS* in *Escherichia coli* showed that the produced protein catalyzes the synthesis of a few sesquiterpenoids, with the alcohol viridiflorol being the main product. To investigate the role of *SiTPS* in the plant-endophyte interaction, an *SiTPS*-over-expressing mutant line was created and assessed for its ability to colonize tomato roots. Although overexpression of *SiTPS* did not lead to improved fungal colonization ability, an in vitro growth-inhibition assay showed that viridiflorol has antifungal properties. Addition of viridiflorol to the culture medium inhibited the germination of spores from a phytopathogenic fungus, indicating that *SiTPS* and its products could provide *S. indica* with a competitive advantage over other plant-associated fungi during root colonization.

## 1. Introduction

Apart from their role in the general metabolism, terpenoids comprise a large and structurally diverse group of specialized metabolites. Several terpenoids produced by plants, bacteria or fungi are of value to the pharmaceutical, cosmetic and food industries, while in nature these compounds mainly serve as defense and signaling molecules [1]. Fungi, with an estimated diversity of more than five million species [2], are considered to be a massive, yet untapped source of terpenoids. Biosynthesis of fungal terpenoids has been described more systematically in the phylum Ascomycota (representing around 60% of the described fungal species), while metabolic pathways in basidiomycete fungi (including mushroom-forming fungi) remain obscure. However, constant progress in genome sequencing and comparative annotation is shedding light on the terpenoid biosynthetic mechanisms used by basidiomycetes [3].

Sesquiterpenoid (C15) biosynthesis in Basidiomycota has received increasing attention over the last decade [4,5,6,7,8]. The specific compounds are characterized by rare structures, unusually enhanced antibiotic and cytotoxic activity [9]. The first committed step in their biosynthesis is the formation of the terpenoid scaffold by the cyclization of the linear precursor farnesyl diphosphate (*E,E*-FPP, C15), a reaction catalyzed by enzymes termed sesquiterpene synthases (STSs) [10]. Initially, STSs cleave off the diphosphate moiety from *E,E*-FPP, forming a reactive carbocation, which undergoes a series of cyclizations and rearrangements. These enzymes stabilize the carbocation intermediates, until the cascade is terminated by proton abstraction, or quenching by water. Alternatively, STSs can use a secondary carbocation, formed from an *E,E*-FPP isomer, the (3R)-nerolidyl diphosphate (*3R*-NPP), and then proceed to the formation of the terpenoid skeleton (Figure 1). 

Interestingly, a comprehensive phylogenetic analysis of all available basidiomycete STSs showed that these enzymes are grouped in phylogenetic clades (phylogenetic clusters), not only according to sequence similarity, but also based on their cyclization mechanism [11]. In detail, all the enzymes of each phylogenetic clade (Figure 2) catalyze the same first carbon–carbon (C–C) bond forming reaction (e.g., Clade II enzymes perform 1,10 cyclization of (3R)-NPP) (Figure 1) [3]. This means that a simple phylogenetic analysis can offer a predictive framework for targeted discovery of sesquiterpenoids with a specific skeleton [11].

*S. indica* (formerly known as *Piriformospora indica*) is an endophytic fungus, belonging to the phylum Basidiomycota and extensively used in plant-fungal interaction studies [16,17,18]. The endophyte can successfully colonize several plant species as a root mutualist [19], conferring several benefits to the host plant, including increased plant biomass [20,21,22,23], and improved plant resistance against biotic [24,25,26,27] and abiotic challenges [20,28,29]. *S. indica* has a characteristic biphasic colonization style, during which a transition from an initial biotrophic growth to a later plant cell-death associated phase is observed [26]. To colonize plant roots successfully, *S. indica* suppresses host immune responses by secreting effectors and interfering with the plant hormone biosynthesis and signaling [26,30,31]. In addition, the endophyte is able to produce phytohormone analogues that contribute further to the establishment of the mutualistic interaction with the host [32,33]. Colonization by *S. indica* was also shown to stimulate host specialized metabolism in several plants [34,35,36], but until now there has been no evidence that this endophytic fungus can itself produce any type of specialized metabolite. In fact, sequencing of the *S. indica* genome showed that the endophyte lacks the large repertoire of terpenoid biosynthetic genes that characterize other basidiomycetes [9]. 

Here, we describe the discovery and the functional characterization of a terpene synthase gene (*SiTPS*), found in the genome of *S. indica*. A comparative phylogenetic analysis showed that SiTPS belongs to Clade I of STSs, together with enzymes that catalyze 1–10 cyclization of *E,E*-FPP. Functional characterization in in vitro and in vivo *Escherichia coli* expression systems showed that SiTPS synthesizes a couple of sesquiterpenes, with viridiflorol, a sesquiterpene alcohol, being the main product. Since it was not shown previously that *S. indica* has the potential to produce terpenoids, we explored the role of SiTPS and its products in the interaction with the host plant and other fungal competitors. We generated an *SiTPS*-overexpressing transgenic *S. indica* line and compared it to a control strain and the wild type *S. indica* with respect to its ability to colonize tomato seedlings. In addition, we performed a growth assay, where spores from the phytopathogenic fungus *Colletotrichum truncatum* were used to inoculate culturing media containing viridiflorol. Overexpression of *SiTPS* did not lead to enhanced colonization ability but the growth assay showed that viridiflorol can inhibit spore germination, indicating that SiTPS could be involved in a *S. indica* defense mechanism. Unravelling the role of SiTPS and the produced compounds during the association between *S. indica*, tomato plants and other plant-associated fungi may provide insights into the role of microbial terpenoids during plant-fungal, fungal-fungal and the tripartite interaction.

## 2. Materials and Methods

### 2.1. Fungal and Plant Material

*S. indica* (isolate DSM11827) was grown at 28 °C in liquid complete medium (CM) [37], supplemented with 2% glucose (*w/v*) on a shaker at 150 rpm, or on solid CM plates, supplemented with 2% glucose (*w/v*) and 1.5% (*w/v*) agar.

Tomato seeds (*Solanum lycopersicum*, cv. Moneymaker) were surface sterilized with 70% ethanol for 1 min, 1% NaClO (*v/v*) for 10 min and rinsed with sterile MilliQ water. The seeds were kept on sterile filter paper in a growth chamber (12 h day 22 °C/12 h night 18 °C, 120 μE/m^2^ s light intensity, 60% relative humidity) until fully germinated (11 days). Plant root inoculation with *S. indica* wild type and mutants was performed by incubating the tomato seedlings in 40 mL of fungal inoculum (300,000 chlamydospores/mL) overnight, on a shaker (120 rpm) at room temperature. For the control treatment, sterile water was used instead of the fungal chlamydospore suspension. After inoculation, the seedlings were sown on Murashige-Skoog (MS) Basal medium (Sigma-Aldrich, St. Louis, MO, USA), supplemented with 1.5% (*w/v*) agar and grown in the same growth chamber until harvest.

### 2.2. Discovery of SiTPS and Phylogenetic Analysis of a Putative Terpene Synthase

The genome of *S. indica* (available at Joint Genome Institute-JGI, https://genome.jgi.doe.gov/Pirin1/Pirin1.home.html; accessed on 15 June 2021) possesses one gene (JGI mRNA:PIIN_06735, called SiTPS from now on) annotated as member of the terpene synthase C family (enzymes that synthesize terpenes of the general and specialized metabolism). The predicted amino acid sequence of SiTPS (JGI Protein Id: 77541) was aligned with Basidiomycota STSs, including the functionally characterized *Coprinopsis cinerea* (Cop1-6) [4,5,6] and *Omphalotus olearius* TPSs (Omp1-10) [3], using the online tool GUIDANCE2 Server (http://guidance.tau.ac.il/ver2/; accessed on 15 June 2021) and the multiple sequence alignment (MSA) algorithm CLUSTALW. A Maximum-likelihood phylogenetic tree was constructed in MEGAX [38] (bootstrap of 500) using the default settings (Figure 2).

The SiTPS sequence was also used in a BLASTp search (https://blast.ncbi.nlm.nih.gov/Blast.cgi; accessed on 15 June 2021) in order to identify similar proteins and investigate its functional context. A selection of proteins (threshold of E-value 2e-67) together with all the putative TPSs from other *Serendipita* species was aligned with SiTPS using again GUIDANCE2 (http://guidance.tau.ac.il/ver2/; accessed on 15 June 2021) (MSA algorithm MAFFT). The alignment file was imported in MEGAX [38] and a maximum-likelihood phylogenetic tree (bootstrap of 500) was constructed using the default settings (Appendix A).

The genomes of *S. indica* and *S. vermifera* were mined for specialized metabolism related genes by performing an antibiotics and Secondary Metabolites Analysis SHell (antiSMASH-fungal version) using the default settings (Appendix A) [39].

### 2.3. SiTPS Expression in Planta and In Vitro

Tomato seedlings were harvested at 11 dpi (days post inoculation) and roots from seven plants were pooled together in each replication. Before RNA extraction, the samples were freeze-dried overnight. Fungal biomass from mature *S. indica* cultures, grown for two weeks on CM agar plates, was also harvested. Total RNA from root and fungal samples was extracted using the Spectrum™ Plant Total RNA Kit (Sigma-Aldrich, St. Louis, MO, USA) and its integrity was validated with an agarose gel 1% (*w/v*). For the plant root samples 1.5 μg of total RNA was used to synthesize cDNA, while for the fungal samples, cDNA was synthesized using 250 ng total RNA. cDNA synthesis for root and fungal samples was performed with the Revert First Strand cDNA Synthesis kit and an oligo-dT primer (Thermo Fisher Scientific, Waltham, MA, USA) with a final volume of 20 μL.

For the RT-qPCR, 2 μL of 1/10 diluted cDNA were used as a template in a 10 μL reaction, using 5 μL of the Brilliant III Ultra-Fast SYBR^®^ Green (Agilent Technologies, Santa Clara, CA, USA) and SiTPS-specific primers (Appendix A). Each RT-qPCR reaction was performed with three technical replications for the four biological replications. RT-qPCR was performed using the AriaMx Real-Time PCR System-G8830A (Agilent Technologies, USA), with a program of 95 °C for 5 min, followed by 40 cycles of 95 °C for 30 s, 60 °C for 1 min and 72 °C for 1 min. A final dissociation step was performed to assess the quality of amplified products and the specificity of the primers. Expression of *SiTPS* was normalized using *S. indica* glyceraldehyde-3-phosphate dehydrogenase gene (*GAPDH*, GenBank: FJ810523.1) expression levels and calculated with the 2^-ΔCt^ method [40,41]. 

### 2.4. Heterologous Expression in E. coli and In Vitro Characterization of SiTPS

The coding region of *SiTPS* (PIIN_06735) was amplified from *S. indica* cDNA (Appendix A) and cloned in-frame with a His-tag (C-terminal) into the NcoI digested pET28b+ vector (Novagen), using the In-Fusion^®^ HD Cloning Kit (Takara Bio,Kusatsu, Japan), according to the manufacturer’s instructions. The resulting plasmid pET_SiTPS was transformed into *E. coli* C41 OverExpress™ cells (Lucigen, Middleton, WI, USA). Heterologous expression and protein purification was performed as described in [42]. Briefly, 500 μL of an overnight culture (5 mL LB broth containing 50 µg/mL kanamycin) was used to inoculate 50 mL of production medium (Terrific Broth medium (pH 7.0), containing 50 µg/mL kanamycin). Cultures were grown at 37 °C in a shaking incubator (180 rpm). When culture OD_600_ reached 0.6, 100 μL of IPTG (Isopropyl β-d-1-thiogalactopyranoside, 0.2 mM) were added to induce expression. Protein expression proceeded overnight at 16 °C in a shaking incubator (180 rpm). The following day, cells were harvested by centrifugation (4500 rpm) at 4 °C for 20 min and lysed using the CelLytic B Cell Lysis Reagent (Sigma-Aldrich, St. Louis, MO, USA), supplemented with 0.1 mg/mL lysozyme, 10 µL/mL protease inhibitor cocktail (Sigma-Aldrich, St. Louis, MO, USA), 0.2 mg/mL benzoase, 25 mM imidazole, 500 mM NaCl and 5% (*v/v*) glycerol. The cell lysate was further used for protein purification, using the His SpinTrap Kit (GE Healthcare, Chicago, IL, USA). Proteins were desalted with the PD MiniTrap G-25 desalting columns (GE Healthcare, Chicago, IL, USA) according to the manufacturer’s instructions and eluted with 600 μL desalting buffer (20 mM HEPES (pH 7.2), 1 mM MgCl_2_, 350 mM NaCl, 5 mM DTT, and 5% (*v/v*) glycerol).

The in vitro terpene synthase assay was performed in a 500 µL reaction that contained 5 µg substrate (GPP, *E,E*-FPP or GGPP (Cayman Chemicals, Ann Arbor, MI, USA)), 100 µg purified enzyme, 10 mM MgCl_2_, 100 mM KCl, 5 mM DTT and 10% *v/v* glycerol in 50 mM HEPES (pH 7.2). The reaction was overlaid with 500 µL n-hexane. Reactions were carried out at 30 °C for 1 h, followed by vortexing to extract the products into the organic phase. Layers were separated by centrifugation and hexane was removed for GC-MS analysis.

### 2.5. In Vivo Characterization of SiTPS

The *in vivo* system was established by modifying the previously developed diterpene production system [43]. The GGPP synthase gene was removed from the pGG vector [44] and replaced by a *Gallus gallus* FPP synthase gene (*FPPS*-Genbank XM_01529864). In detail, the *G. gallus FPPS* was synthesized (Integrated DNA Technologies, Coralville, IA, USA) and cloned into NdeI and XhoI digested pACYCDuet vector (Novagen), using the In-Fusion^®^ HD Cloning Kit (Takara Bio, Kusatsu, Japan). The resulting plasmid pACYCDuet_GgFFPS, together with pIRS, a plasmid containing three upstream genes of the MEP pathway [43], was introduced to *E. coli* C41 OverExpress™ cells (Lucigen, Middleton, WI, USA) to create a farnesyl diphosphate-producing strain. Transformed *E. coli* cells were grown on LB agar plates containing kanamycin (25 µg/mL), chloramphenicol (20 µg/mL), and streptomycin (25 µg/mL) and were verified by colony PCR. A single PCR-positive clone was grown in 50 mL Terrific Broth medium (pH 7.0), with the appropriate antibiotics at 37 °C until the culture reached an OD_600_ of 0.6. The temperature was lowered to 16 °C for 1 h before expression was induced with 1 mM IPTG. The culture was also supplemented with 40 mM pyruvate, 1 mM MgCl_2_ and was grown for an additional 72 h. Cell pellets were extracted in 50 mL of n-hexane, the organic phase was concentrated under N_2_ and analyzed by GC-MS.

### 2.6. Gas Chromatography-Mass Spectrometry (GC-MS) Analysis

GC-MS analysis was performed as described by [45] on an Agilent 7890A GC with an Agilent VF-5ms column (30 m × 250 µm × 0.25 µm, with 10 m EZ-Guard) and an Agilent 5975C detector. The inlet was set to 250 °C splitless injection of 1 µL helium carrier gas with a column flow of 1 mL/min. The detector was activated after a three-minute solvent delay. The oven temperature ramp started at 80 °C and held for 1 min, then increased to 130 °C by 40 °C/min, increased to 250 °C by 10 °C/min, following by a final increase of 100 °C/min to 325 °C, where it was held for 3 min. Obtained spectra were compared with NIST17 Mass Spectral Database. Analytical standards of viridiflorol (CAS 552-02-3) and ledol (CAS 577-27-5) were purchased from Sigma Aldrich (Cat No. 72999-10MG) and Santa Cruz (Cat No. sc-396548), respectively. 

### 2.7. Construction of Overexpression Plasmids and S. indica Peg-Mediated Transformation 

For creating a *SiTPS*-overexpressing mutant, the *SiTPS* coding sequence was cloned into a NheI and PmeI digested K167 vector, a *S. indica*-compatible vector developed in the Zuccaro lab (S. Wawra and H. Widmer, unpublished; modified from [46]) using the In-Fusion^®^ HD Cloning Kit (Takara Bio, Kusatsu, Japan). The resulting plasmid K167_SiTPSov carried the *SiTPS* coding sequence under the control of a strong promoter (FGB1 promoter-Fungal Glucan-Binding 1, PIIN_03211) (Appendix A). The empty vector named K167_ev was transformed as control.

*S. indica* protoplasts were isolated and transformed with K167_SiTPSov and K167_ev through polyethylene glycol (PEG)-mediated transformation according to [47]. Young mycelium from a 7-day-old *S. indica* culture was harvested, homogenized and left to regenerate for three further days. The regenerated mycelium was treated with 20 mL SMC solution (1.33 M sorbitol, 50 mM CaCl_2_ and 20 mM MES) containing 0.02 g/mL lysing enzyme from *Trichoderma harzianum* (Sigma-Aldrich, St. Louis, MO, USA) at 32 °C. After 2 h, protoplasts were harvested and transformed with 7–10 μg linearized and purified plasmid in the presence of 40% PEG 3350 and heparin (15 mg/mL). Protoplast were regenerated on plates with two layers of malt yeast peptone (MYP) agar supplemented with 0.3 M sucrose (0.7% *w/v* malt extract, 0.1% *w/v* peptone, 0.05% *w/v* yeast extract). The bottom medium contained 1.2% agar and hygromycin B (80 μg/mL). The top medium (0.6% agar and no antibiotics) was mixed with the transformation mixture and then quickly poured on to the solidified bottom medium. The plates were incubated at 28 °C. After 10–14 days, regenerated colonies were transferred to CM plates supplemented with 80 μg/mL hygromycin B and verified for carrying either K167_SiTPS or K167_EV. *S. indica* mutants that were not impaired in growth and with the right mating type were selected and used for further experiments (Appendix A). 

### 2.8. Quantification of Root Colonization by S. indica and Mutants with qPCR

Genomic DNA was extracted from root samples of inoculated tomato plants at 2 and 11 dpi using the DNeasy^®^ Plant Mini Kit (Qiagen, Hilden, Germany). Roots from eight plants were pooled in one replication and three replications were included per time-point and treatment. The qPCR was performed in the AriaMx Real-Time PCR System-G8830A (Agilent Technologies, Santa Clara, CA, USA) using a cycling program of 95 °C for 5 min, 30 cycles of 95 °C for 30 s, 62 °C for 1 min, and 72 °C for 1 min. A final dissociation step was performed to assess the quality of amplified products and the specificity of the primers. In a 10 μL reaction, 5 μL Brilliant III Ultra-Fast SYBR^®^ Green (Agilent Technologies, Santa Clara, CA, USA), 1 μL of total DNA (30–50 ng/μL) and 0.4 μM of plant- or fungus-specific primers were used (Appendix A). 

Fungal colonization was determined using the ratio of fungal to plant DNA. For the quantification of fungal DNA, a standard curve was generated using serial dilutions of DNA from a pure *S. indica* culture and specific primers for the Internal Transcribed Spacer-ITS (*SiITS*, GenBank: NR_119580.1, *S. indica* wt: *y* = –3.2357*x* + 7.599 and R^2^ = 0.9969, *S. indica* evV: *y* = –3.3687*x* + 9.3553 and R^2^ = 0.9942, *S. indica* ovII: *y* = –3.3667*x* + 10.845 and R^2^ = 0.9954). Plant DNA was quantified similarly, using a standard curve of plant DNA dilutions and specific primers for *S. lycopersicum* β-tubulin gene (*SlTUB*, GenBank: DQ205342.1) (*y* = –3.3527*x* + 16.513 and R^2^ = 0.999). 

### 2.9. Testing the Antimicrobial Activity of Viridiflorol

The antifungal activity of viridiflorol was determined using a growth assay, where the plant pathogenic fungus *C. truncatum* (syn. *Colletotrichum capsici*) was grown in liquid medium containing different concentrations of the compound. Viridiflorol, purchased by BOC Sciences (NY, USA), was initially diluted in methanol to a stock solution of 10 mg/mL. The stock solution was used to prepare the working solutions, which contained either viridiflorol in concentrations of 500, 1000, 2000 or 5000 µg/mL, or methanol in the respective concentrations that served as control. In the wells of a 96-well microtiter plate, 25 µL of each working solution and 225 µL Potato Dextrose Broth (PDB) were added, ending up to the test concentrations of 50, 100, 200 or 500 µg/mL viridiflorol or methanol. Hygromycin (400 µg/mL) was added in some wells instead of viridiflorol or methanol, as an antifungal drug that inhibits growth of the respective pathogens. The wells were inoculated with spores from *C. truncatum* isolate CP2304 (4000 spores) from the fungal collection in Copenhagen. The plate was incubated on a shaker (100 rpm) for 6 days at room temperature and pictures were taken to identify fungal growth. Each treatment was performed in four replicates and the whole assay was done twice. 

### 2.10. Statistical Analysis

The data from *SiTPS* relative expression was subjected to a *t*-test in order to compare pairwise gene expression levels between the two different growth conditions of *S. indica* (in tomato roots and on synthetic medium). Colonization rates of different *S. indica* strains represent continuous variables and were analyzed by analysis of variance, assuming a normal distribution (one-way ANOVA). For all the statistical tests performed, hypotheses were rejected at *p* < 0.05 and all data were analyzed in R-Studio (R version 3.4.1, https://cran.r-project.org/bin/windows/base/old/3.4.1/). 

## 3. Results

### 3.1. SiTPS Belongs to Clade I of Basidiomycota STSs

A phylogenetic tree containing previously characterized basidiomycete STSs (SiTPS included) [3,4,6,8,12,14] was constructed and it showed that the specific enzymes are grouped into four distinct STS clades (Figure 2), also observed previously [3,11]. SiTPS was found in Clade I, which includes enzymes that catalyze formation of sesquiterpenoids through an initial C1-C10 closure of *E,E*-FPP (Figure 1).

An additional phylogenetic analysis was also conducted and included a selection of proteins, closely related to the *S. indica* STS (Appendix A). The specific tree showed that SiTPS was grouped together with putative terpene synthases from other *Serendipita* species. In addition, the *Serendipita* sequences displayed close relation to a protein belonging to the lichen *Cladonia uncialis*.

### 3.2. SiTPS Is Induced during Root-Colonization

Expression of *SiTPS* was studied under two different growth conditions. Relative expression of *SiTPS* was quantified in fungal biomass from 14-day-old *S. indica* colonies, grown on complete medium (CM) agar plates and compared to expression in *S. indica*-colonized tomato roots (11 days post inoculation, dpi) (Figure 3). Gene expression was significantly upregulated by 3-fold when *S. indica* was growing *in planta* in comparison to the control samples.

### 3.3. SiTPS Encodes for a Viridiflorol Synthase

The biochemical activity of SiTPS was elucidated by using an in vitro and in vivo heterologous expression system (Figure 4). In the in vitro assays, the purified SiTPS, heterologously produced in *E. coli* cultures, was incubated with three terpene substrates (geranyl diphosphate- GPP, *E,E*-FPP and geranylgeranyl diphosphate- *E,E,E*-GGPP). Hexane extracts showed activity only when SiTPS was incubated with *E,E*-FPP (Appendix A).

To validate results from the in vitro assays, an *E. coli*-based in vivo system was used as well. GC-MS analysis of hexane extracts from the bacterial culture co-expressing *SiTPS* with genes necessary for *E,E*-FPP production showed a number of peaks that were absent in the extract derived from the *E,E*-FPP-producing culture without SiTPS. The main product, matching the product from the in vitro assays in retention time and mass spectrum, showed closest matches to the sesquiterpene isomers viridiflorol and ledol in the NIST17 mass spectra database. Comparison with the authentic standard identified the product as viridiflorol (Figure 4). Based on the phylogenetic position of SiTPS (Figure 2) and the functional characterization of the enzyme (Figure 4), a putative cyclization mechanism for viridiflorol by SiTPS was proposed (Figure 5).

### 3.4. Role of SiTPS during S. indica-Colonization of Tomato Roots

Even though *SiTPS* was shown to express a functional viridiflorol synthase in *E. coli* expression systems, no terpenoids have been detected previously in *S. indica* cultures [48]. In addition, data showing *SiTPS* upregulation in fungal tissues associated with tomato roots (Figure 3) suggested that SiTPS products might play a role during plant colonization. To investigate the role of SiTPS in root colonization, *S. indica* mutants over-expressing *SiTPS* were designed. According to the antiSMASH analysis performed on the *S. indica* genome, SiTPS is a stand-alone gene and none of the genes in genomic proximity is related to terpene biosynthesis (Appendix A). Based on that, we hypothesized that overexpressing *SiTPS* would have an accumulative effect on the amount of compounds produced.

After protoplast transformation, the regenerative clones were screened for having all four mating type genes (heterokaryotes) (Appendix A) and one *SiTPS*-overexpressing transformant (ovII) was selected and checked for *SiTPS* expression (Appendix A). The ovII strain was used in time-course colonization experiments to evaluate its ability to colonize plant roots. An *S. indica*-transformant carrying the empty vector (control treatment-evV) and the wild type *S. indica* were also used in this colonization study. 

The ratio of fungal DNA to plant DNA was used as a measurement of the colonization ability of each *S. indica* strain. The colonization ratio was estimated in colonized roots at 2 and 11 dpi, but no difference was observed in the colonization ability between the over-expressing and the empty vector-carrying transformants at the selected time-points. No difference was observed in colonization ability between *S. indica* transformants and the wild type either (Table 1).

### 3.5. Viridiflorol Has Antifungal Properties

To study whether SiTPS is involved in a fungal defense mechanism, we performed an in vitro growth assay using viridiflorol, the main terpene produced by SiTPS. Spores from the plant pathogen *Colletotrichum truncatum* (syn. *Colletotrichum capsici*) isolate CP2304 were added in the wells of a microtiter plate, containing liquid medium supplemented with viridiflorol in different concentrations. The compound inhibited germination of spores and consequently growth of *C. truncatum* at a concentration of 100 µg/mL and higher, whereas the respective methanol controls did not affect at all fungal growth (Figure 6). 

## 4. Discussion

### 4.1. S. indica Possesses a Viridiflorol Synthase

The in vitro and in vivo characterization assays showed that SiTPS accepts *E,E*-FPP to produce a mix of sesquiterpenoids with the main product identified as viridiflorol. Viridiflorol is a rare sesquiterpenoid with a 7/5/3 tricyclic scaffold belonging to the group of aromadendranes, which are terpenoids with a characteristic skeleton of a dimethyl cyclopropane ring fused to a hydroazulene ring system [49], usually used in fragrance production [50].

During sesquiterpenoid synthesis, FPP cyclization can occur either with the closure of *E,E*-FPP or its isomer, (*3R*)-NPP. A mechanism for the formation of viridiflorol was previously proposed, based on quantum chemical calculations, starting with the cyclization of *E,E*-FPP by a ring closure of C1 and C10 [51]. Phylogenetic analysis showing that SiTPS belongs to Clade I (Figure 2), together with enzymes that perform 1,10 cyclization of *E,E*-FPP, supports that activity of SiTPS proceeds through route a (Figure 5). On the contrary, another viridiflorol synthase isolated from the black poplar mushroom, *Cyclocybe aegerita* (syn. *Agrocybe aegerita*) [14], named AaVS, was placed in Clade II, together with enzymes that perform 1,10 cyclization of (3R)-NPP, indicating that biosynthesis of viridiflorol can actually occur through both routes (Figure 5).

Viridiflorol has also been detected in extracts of many plant species belonging to the families Myrtaceae [52] and Lamiaceae [53,54], but until now only two viridiflorol synthases of plant origin have been identified [52]. The specific compound has also been encountered as a minor product in fungal extracts [55,56] and after AaVS [14], SiTPS is the second fungal enzyme identified and experimentally characterized as a viridiflorol synthase. 

### 4.2. Overexpression of SiTPS Does Not Affect S. indica Colonization Ability

*SiTPS* was found to be upregulated at 11 dpi in colonized tomato roots, when its expression was compared to that from a mature fungal colony grown on synthetic medium. However, gene expression data generated from a microarray assay performed on *S. indica*-colonized barley roots showed that *SiTPS* expression is induced *in planta* but at different timepoints [57]. In details, when we re-analyzed the data from barley roots, transcripts of *SiTPS* were found to accumulate specifically at the very early stages of colonization, around 36 to 48 h post inoculation (hpi), both in living and dead colonized roots, when compared to growth on synthetic medium. However, a similar expression pattern was not observed during later stages (5 dpi). This observation could possibly suggest that the gene is more important at the very early stages of the interaction, during the pre-penetration phase, or that differences in *SiTPS* expression evaded detection by the specific method. Another observation in this study was that when dead and living barley roots were compared, accumulation of *SiTPS* transcripts was higher in living roots, indicating that *S. indica* activates *SiTPS* when association with living tissues takes place. 

Our *SiTPS* expression data and former literature led us to hypothesize that *SiTPS* can be involved in colonization of plant roots by *S. indica*. Hence, we established a root-colonization assay, where tomato seedlings were inoculated with either an *SiTPS*-overexpressing mutant or control strains and the ratio between fungal to plant DNA was used to evaluate fungal colonization ability. The *SiTPS*-overexpressing mutant appeared to colonize tomato roots to the same extent as the empty vector-carrying transformant and the wild type *S. indica*, showing that overexpression of the *S. indica* viridiflorol synthase gene does not influence fungal colonization ability at the specific time-points and in our experimental set-up. In another study, the colonization ability of *Trichoderma virens*, an endophytic fungus able to produce several volatile terpene compounds, was compared with this of deletion mutants and other *Trichoderma* species that lack the terpene synthase gene responsible for the synthesis of these metabolites [58]. Similar to our findings, the specific study showed that the ability to produce terpenes did not correlate with increased fungal colonization ability, since there was no significant difference between colonization ratio of the fungal strains used.

However, these results do not exclude that the products of *SiTPS* could play another role *in planta*. For example, fungal sesquiterpenoids could act as signaling or defense molecules against other root-residing microorganisms. In fact, according to a whole transcriptome study performed on colonized roots of barley plants, a putative terpene synthase gene from the orchid mycorrhizal fungus *Serendipita vermifera* ssp. *bescii*, was upregulated significantly only when another fungus (the pathogen *Bipolaris sorokiniana*) was colonizing the plant roots [59].

### 4.3. Viridiflorol Could Serve as a Defense Compound against Other Plant-Associated Fungi

It has been shown repeatedly that volatile organic compounds, including volatile terpenoids, mediate communication between plants and microorganisms [60]. Several plant terpenoids, such as α- and β-pinene or abietic acid, have been considered to be positively involved in the development of ectomycorrhizal associations, mainly during the pre-symbiotic phase [61,62]. On the other hand, little is known about fungal terpenoids and their role in endophytism and to our knowledge, there is no direct evidence that they facilitate establishment of plant-endophytic interactions. Terpenoids produced by fungi are considered to be implicated mostly in defense against antagonists or in signaling with other microorganisms [60,63]. Therefore, we redirected the focus of our study into exploring the potential of viridiflorol as a defense compound. The in vitro growth assay showed that high concentrations of viridiflorol (>100 μg/mL) could successfully inhibit the germination and consequently the growth of the phytopathogenic fungus *C. truncatum*. Previous studies have also shown that viridiflorol has weak to moderate anti-inflammatory and antimicrobial activity [49,64,65,66], suggesting that the ability to produce this compound could provide *S. indica* with an advantage when growing in a more competitive environment, where several rhizospheral microorganisms are fighting for a niche within the plant tissue. 

In our case, antagonism assays and co-cultures including *S. indica* wild type or mutants and other plant-associated microorganisms might further support our hypothesis about viridiflorol being implicated in defense. Additionally, *in planta* experiments with knockout *S. indica* mutants and other microbial competitors could provide an even more realistic image regarding the actual role of SiTPS and sesquiterpenes during a tripartite association between a plant, a pathogen, and a beneficial symbiont. Identifying the role of endophytic terpenoids during symbiotic associations will provide a more in-depth understanding of these complex interactions and their effect in the surrounding environment, opening up new opportunities for their application in agriculture.

## 5. Patents

The authors have filed the patent application “Method for producing the sesquiterpene viridiflorol with a fungal enzyme” describing the function of the *S. indica* terpene synthase (U.S. Patent Application Serial No.: 62/899,391).

## Figures and Tables

**Figure 1 biomolecules-11-00898-f001:**
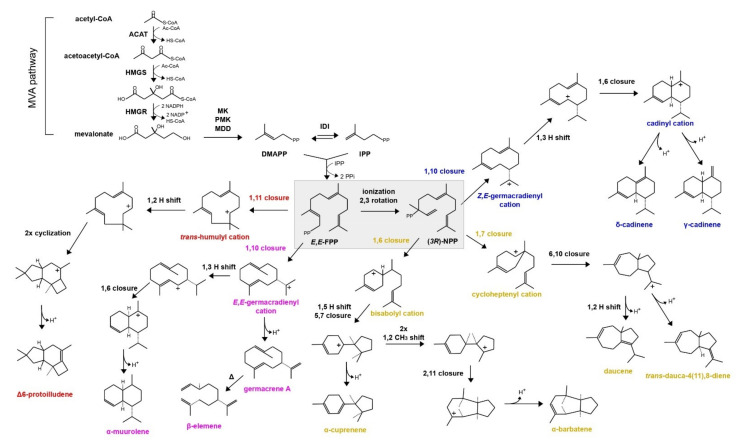
Proposed biosynthetic pathways of several sesquiterpenes found in basidiomycete fungi. Sesquiterpenoid scaffolds (and the respective STSs) can be distinguished by the first cyclization step of *E,E*-FPP or (*3R*)-NPP. Here, four distinct structural groups are presented in different colors, each one containing terpenoids formed by different types of STSs. Terpenoids derived from 1,10 closure of *E,E*-FPP are shown in pink (STS Clade I), 1,11 closure of *E,E*-FPP are shown in red (STS Clade III), 1,6 or 1,7 closure of (*3R*)-NPP are shown in yellow (STS Clade IV) and 1,10 closure of (*3R*)-NPP are shown in blue (STS Clade II) (adapted from [3]).

**Figure 2 biomolecules-11-00898-f002:**
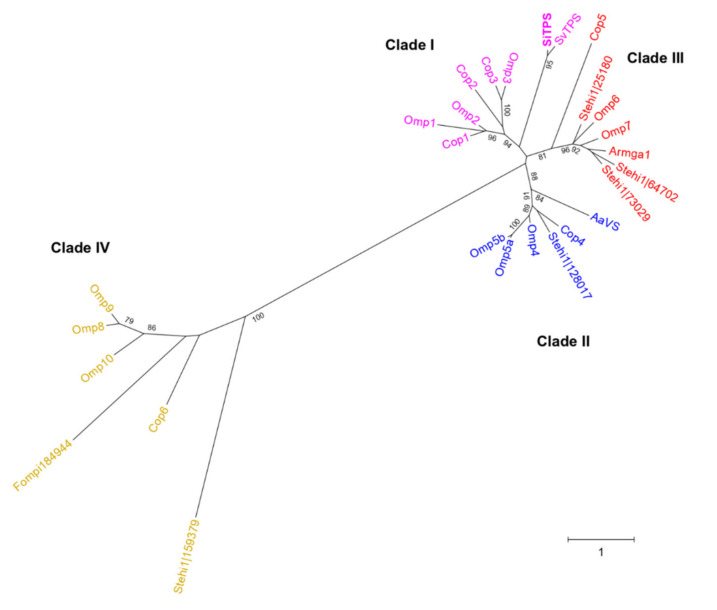
Phylogenetic analysis of selected functionally characterized basidiomycete STSs, Cop1-6 [4,6] from *Coprinopsis cinerea*, Omp1-10 from *Omphalotus olearius*, Fompi|84944 from *Fomitopsis pinicola* [3], Stehi1|159379, 128017, 25180, 64702, 73029 [8,12] from *Stereum hirsutum*, ArmGa1 [13] from *Armillaria gallica*, AaVS [14] from *Cyclocybe aegerita* (syn. *Agrocybe aegerita*), SiTPS from *Serendipita indica* and the SvTPS (NCBI accession number: PVF97777.1) from *Serendipita vermifera* were included in the analysis. A maximum-likelihood tree shows the 4 different STS clades formed, described also in [3,15] and Figure 1. The enzymes falling in one clade are considered to catalyze the first cyclization of the substrate in the same specific way; STS Clade I catalyze 1,10 cyclization of *E,E*-FPP (pink); Clade II, 1,10 cyclization of (*3R*)-NPP (blue); Clade III, 1,11 cyclization of *E,E*-FPP (red); Clade IV, 1,6 or 1,7 cyclization of (*3R*)-NPP (yellow). SiTPS is found in Clade I. The scale bar indicates a genetic distance of 1 and the bootstrap values are shown below the branches.

**Figure 3 biomolecules-11-00898-f003:**
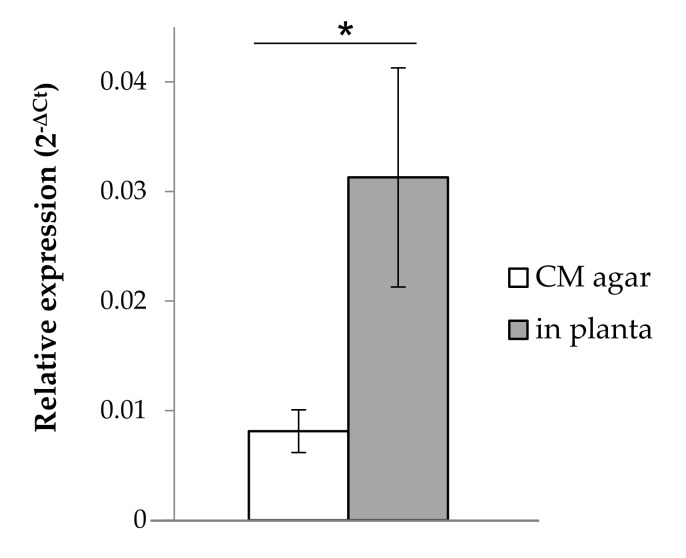
Relative expression of *SiTPS* when *S. indica* is grown for 14 days on complete medium (CM) agar plates compared to *S. indica* colonizing tomato roots (11 days post inoculation, dpi). Relative expression under both conditions was normalized using *SiGAPDH* (GenBank: FJ810523.1) expression levels and calculated with the 2^−ΔCt^ method [39,40]. Error bars represent the standard error of the mean (*n* = 4). The asterisk (*) indicates a significant difference between the two conditions (*t*-test in R-studio, * *p* ≤ 0.05).

**Figure 4 biomolecules-11-00898-f004:**
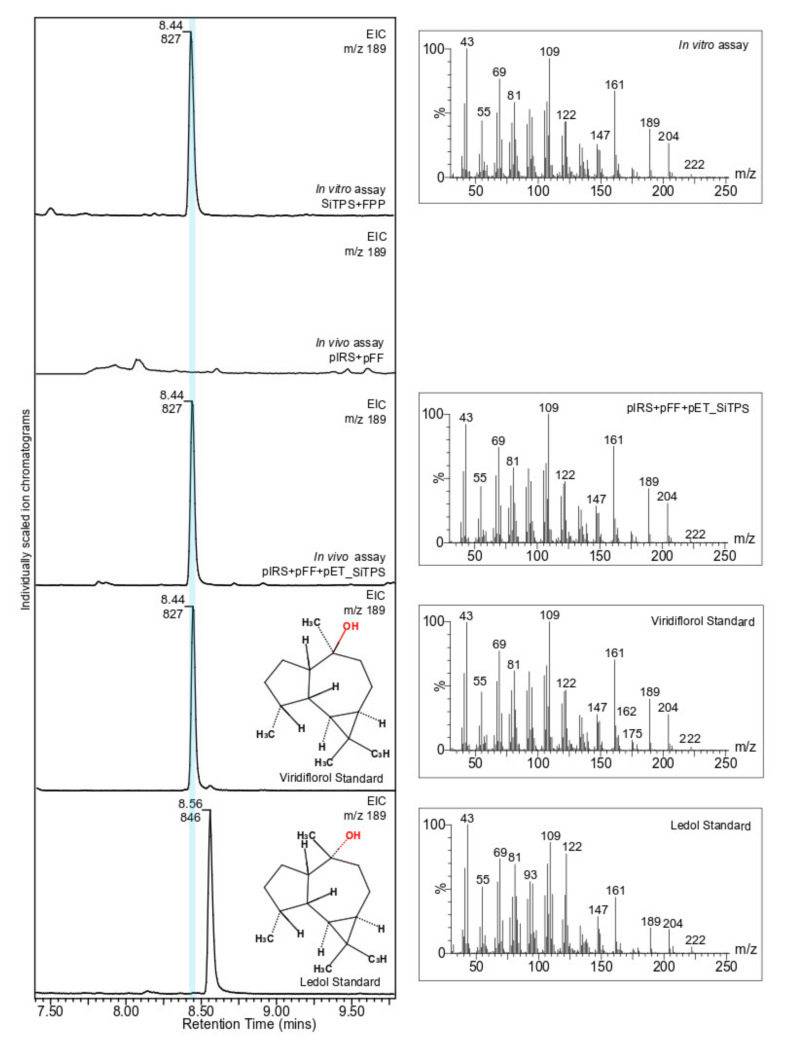
In vivo and in vitro characterization of SiTPS. GC-MS of hexane extracts from the in vitro assay in which the purified SiTPS was incubated with the substrate FPP (*E,E*-FPP) and from the *in vivo* assay with *E. coli* strains transformed with the plasmids pIRS, pFF and pET_SiTPS. An *E. coli* strain carrying only the pIRS and pFF plasmids was used as control. Analytical standards of viridiflorol and ledol were used to identify the produced compound. The mass spectra of the produced compounds and the analytical standards are given in the right hand panels.

**Figure 5 biomolecules-11-00898-f005:**
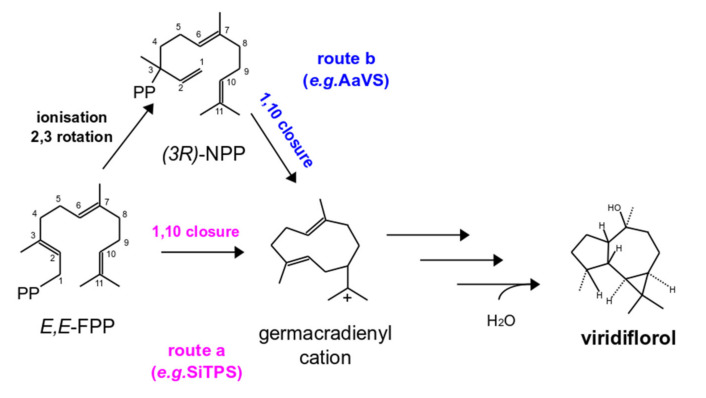
Proposed biosynthetic pathway leading to the production of viridiflorol. Viridiflorol biosynthesis starts with a 1,10-closure of either *E,E*-FPP (route a-Clade I, e.g., SiTPS) or (*3R*)-NPP (route b-Clade II, e.g., AaVS) to form a germacradienyl cation (*E,E* or *Z,E*, respectively). Two cyclization events (e.g., 1,11 and 2,6 closure) and additional reactions (e.g., addition of a molecule of water) result to formation of viridiflorol.

**Figure 6 biomolecules-11-00898-f006:**
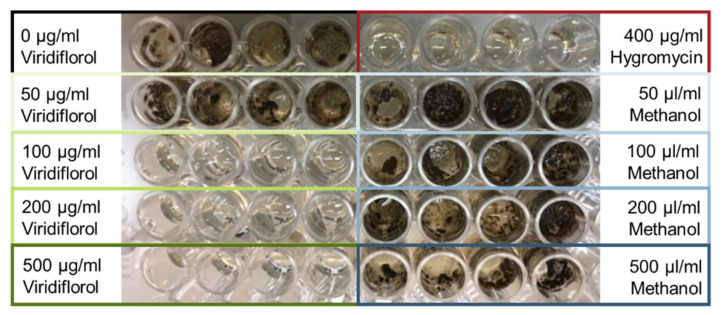
Microtiter plate wells containing PDB supplemented with viridiflorol, methanol or hygromycin and inoculated with 4000 spores of *C. truncatum*. Each treatment was performed in four replicates.

**Table 1 biomolecules-11-00898-t001:** Colonization of tomato roots by *S. indica* wild type (wt), the empty vector-carrying (evV) and the *SiTPS*-overexpressing (ovII) mutants. The colonization ability of each strain was estimated using the ratio of fungal DNA (DNAf) to plant DNA (DNAp) at 2 and 11 days post inoculation (dpi). No statistical differences were observed in between the different treatments (one-way ANOVA in R-studio). Standard error of the mean (SEM) is also presented (*n* = 3).

*S. indica* Strain	2 dpi DNAf/DNAp ± SEM	11 dpi DNAf/DNAp ± SEM
wt	0.0044 ± 0.0003	0.0324 ± 0.0026
evV	0.0052 ± 0.0001	0.0468 ± 0.0102
ovII	0.0054 ± 0.0008	0.0301 ± 0.0040

## Data Availability

Not applicable.

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
