# Peer review of "A Sesquiterpene Synthase from the Endophytic Fungus Serendipita indica Catalyzes Formation of Viridiflorol"

_biomolecules, 2021, doi:10.3390/biom11060898_

Round 1

Reviewer 1 Report

Dear Authors,

This is an interesting study. The content of the paper could be published in Biomolecules. But, the manuscript requires some minor reviews. In the manuscript, I pointed out some correction.

Line 30: remove “Serendipita indica” and add “antifungal”

Line 38: Change “3 to 5 million” to “more than 5 million”

Line 38: Change “characterized by rare structures and unusually enhanced antibiotic and cytotoxic activity” to “characterized by rare structures, unusually enhanced antibiotic and cytotoxic activity”

Line 71: I think it would be better if the sentence “In detail, all the enzymes of one phylogenetic clade” Change to “In detail, all the enzymes of each phylogenetic clade”

Line 90: Change “Serendipita indica” to “S. indica

Line 116: Change “phytopathogenic fungus” to “phytopathogenic fungus Colletotrichum truncatum

Line 126: Change “Serendipita indica” to “S. indica

Line 130: 10 min is too much for seed surface sterilization using 1% NaClO (v/v)!!! It can affect seed germination ability. Please check it again.

Line 163: Change “11 dpi” to “11 dpi (11 days post inoculation)”

Line 285: β-tubulin should be written in italic

Line 290: Change “plant pathogen” to “plant pathonenic fungus C. truncatum (syn. Colletotrichum capsici)

Line 299: Change “Colletotrichum truncatum (syn. Colletotrichum capsici)” to “C. truncatum

Line 300: do you mean 4000 spores/ml??? Please, clarify it.

Line 411-415: transfer this part to materials and methods.

Line 421: do you mean 4000 spores/ml??? Please, clarify it.

Line 499: Change “phytopathogenic fungus” to “phytopathogenic fungus C. truncatum

Reviewer 2 Report

This work describes the discovery and the functional characterization of a terpene synthase gene found in the genome of the endophytic fungus Serendipita indica, which synthesizes the sesquiterpene viridiflorol. The authors showed that this compound could successfully inhibit the growth of a phytopathogenic fungus, which could provide S. indica with a competitive advantage over other fungi within the plant tissue.

The overall scientific content is high. It is a very interesting study, which is well done and well written. So, in my opinion the paper only requires some minor modifications to make it suitable for publication.

1.References should be checked:

- Ref. 19 and 48 should include DOI numbers

- Ref. 22, 23, 38, 45, 48, 54 and 66 should be completed (volume and/or pages)
